# Diversity and Function of Appressoria

**DOI:** 10.3390/pathogens10060746

**Published:** 2021-06-12

**Authors:** K. W. Thilini Chethana, Ruvishika S. Jayawardena, Yi-Jyun Chen, Sirinapa Konta, Saowaluck Tibpromma, Pranami D. Abeywickrama, Deecksha Gomdola, Abhaya Balasuriya, Jianping Xu, Saisamorn Lumyong, Kevin D. Hyde

**Affiliations:** 1Innovative Institute of Plant Health, Zhongkai University of Agriculture and Engineering, Guangzhou 510225, China; tchethi@yahoo.com; 2Center of Excellence in Fungal Research, Mae Fah Luang University, Chiang Rai 57100, Thailand; ruvi.jaya@yahoo.com (R.S.J.); yui2134000@gmail.com (Y.-J.C.); sirinapakonta@gmail.com (S.K.); pranamiabeywickrama@yahoo.com (P.D.A.); deeckshagomdola@gmail.com (D.G.); 3School of Science, Mae Fah Luang University, Chiang Rai 57100, Thailand; 4CAS Key Laboratory for Plant Diversity and Biogeography of East Asia, Kunming Institute of Botany, Chinese Academy of Sciences, Kunming 650201, China; saowaluckfai@gmail.com; 5Beijing Key Laboratory of Environment Friendly Management on Diseases and Pests of North China Fruits, Institute of Plant and Environment Protection, Beijing Academy of Agriculture and Forestry Sciences, Beijing 100097, China; 6Department of Plant Sciences, Faculty of Agriculture, Rajarata University of Sri Lanka, Mihintale 50300, Sri Lanka; abhayab2006@gmail.com; 7Department of Biology, McMaster University, 1280 Main Street West, Hamilton, ON L8S 4K1, Canada; jpxu@mcmaster.ca; 8Center of Excellence in Microbial Diversity and Sustainable Utilization, Faculty of Science, Chiang Mai University, Chiang Mai 50200, Thailand; scboi009@gmail.com; 9Academy of Science, The Royal Society of Thailand, Bangkok 10300, Thailand

**Keywords:** host-defense, host-recognition, hyaline appressoria, infection process, melanized appressoria, proto-appressoria

## Abstract

Endophytic, saprobic, and pathogenic fungi have evolved elaborate strategies to obtain nutrients from plants. Among the diverse plant-fungi interactions, the most crucial event is the attachment and penetration of the plant surface. Appressoria, specialized infection structures, have been evolved to facilitate this purpose. In this review, we describe the diversity of these appressoria and classify them into two main groups: single-celled appressoria (proto-appressoria, hyaline appressoria, melanized (dark) appressoria) and compound appressoria. The ultrastructure of appressoria, their initiation, their formation, and their function in fungi are discussed. We reviewed the molecular mechanisms regulating the formation and function of appressoria, their strategies to evade host defenses, and the related genomics and transcriptomics. The current review provides a foundation for comprehensive studies regarding their evolution and diversity in different fungal groups.

## 1. Introduction

Host-fungal interactions found in nature are described as commensalism, symbiosis, or pathogenesis. For any successful fungal-host interaction, the most crucial steps are attachment, lysis, and mechanical breaching of the host surface [1,2]. To overcome the barriers present in the host, fungi have evolved diverse strategies such as the formation of specialized infection structures and the secretion of cell-wall-degrading enzymes, toxins, and effectors [1,3,4,5,6]. Infection structures which are believed to have evolved from non-pathogenic hyphae [1] are variable and belong to many types such as appressoria, expressoria, infection cushions, infection hyphae, and penetration hyphae [1,7,8].

Appressoria, discovered by Frank [9] from the pathogenic fungus *Colletotrichum lindemuthanium* as the ‘spore-like organ’ or the ‘adhesion organ’, represent the most studied form of infection structure. Different types of appressoria and their mechanisms of penetration have been studied extensively in plant pathogens, and also in endophytes, saprobes, other symbionts (including arbuscular mycorrhizal fungi and lichens), and entomopathogenic fungi [3,4,6,8,10]. Highly specialized, melanized appressoria with extremely high turgor pressure, which is critical for cuticle penetration, have been identified from *Pyricularia grisea* (*Pyr.*), *Pyr. oryzae*, and *Colletotrichum* (*Col*.) species [1,4,8]. Non- or lightly-melanized appressoria have been identified from *Blumeria graminis* (*Bl.*), *Ustilago maydis*, and *Cochliobolus* (*Coc.*) species [1,11]. Similarly, infection cushions have been identified from *Fusarium graminearum*, *Rhizoctonia* sp., *Sclerotium* sp., *Sclerotinia* sp. (*Sc.*), and *Botrytis cinerea* (*Bot.*) [12,13,14]. Furthermore, different appressoria have been observed in entomopathogenic *Beauveria bassiana* (*Be.*), mutualistic arbuscular mycorrhizae in Glomeromycota, and the oomycetes such as *Phytophthora* (*Ph.*), *Pythium* (*Pyt.*), and *Peronospora* species [10,15]. For example, oomycete appressoria are non-melanized or non-pigmented and smaller compared to fungal appressoria, and they are separated from germ tubes by false septa, while in fungi a septum separates the appressorium from the spore [16].

Adherence of the fungal spore to the host plant is considered as a critical requirement for spore germination, germ tube elongation, and appressorium formation during the plant-fungal interactions [2]. These morphogenetic events from spore attachment to appressorium formation are stimulated by a variety of host plant signals and environmental factors such as substratum hydrophobicity [17], topographic signals [18], cutin monomers [19], and ethylene signals [2]. Several important fungal signaling pathways have been heavily studied for their involvement in the formation and functioning of appressoria, including G proteins, cAMP, MAP kinases, and calcium/calmodulin-mediated signaling pathways [20,21]. Many comprehensive studies have been conducted on diverse fungal invasion strategies, and the morphology of these processes are well-documented [22,23,24,25,26,27].

Genomics and their interconnected disciplines have provided a fertile platform to address most of these issues and controversies related to the formation and functioning of appressoria. Genomic data and novel techniques have contributed towards a better understanding of these fungal-host interactions from appressorium formation to host invasion [28,29,30,31,32]. Advances in reverse genetic techniques have enabled cloning of diverse genes predicted to be involved in these interactions, and many are characterized via various functional genomic approaches [17,21,28,33,34,35,36,37,38,39,40,41,42].

This study brings together diverse information on appressoria. The review explains what appressoria are, the genera that produce appressoria, how appressoria are involved in host colonization, the ultrastructure of appressorial colonization, and their latest molecular understandings. Further, recent discoveries from omics studies including genomics, transcriptomics, and proteomics related to appressorial formation and function are discussed. This knowledge is important to understand the appressorial diversity and evolution among different fungal life strategies.

## 2. What Are Appressoria?

Appressoria have been defined in a number of ways, often depending on the discipline of the definer. The commonly used definitions are given in Table 1. Most definitions refer to appressoria as being infection structures of pathogens. Fungal appressoria are, however, also found in epiphytes, endophytes, and even saprobes. Many entomo-pathogenic fungi also develop appressoria [10,41]. We, therefore, propose to define appressoria in a more general sense.

### General Definition

Appressoria are specialized cells or adhesion structures produced by fungi from which a penetration peg emerges that pierces or enters the host tissues.

The term ‘appressoria’ was first introduced by Frank when he studied diseases of poplar trees and beans caused by *Fusicladium* sp. and *Col. lindemuthianum*, respectively [9]. He described them as spore-like organs, formed at the end of germ tubes. The term was later coined as the ‘adhesion organ’ when he found that appressoria are tightly attached to the host surface. Hasselbring confirmed Frank’s findings [43]. In Hasselbring’s experiments, he placed the spores of *Gloesporium fructigenum* in a drop of water on a glass slide to study appressoria formation. Within 12 to 18 hours, he found that the spores which touched the surface of the glass generated short germ tubes; later, the apices of the germ tubes became enlarged and formed thick-walled, brown spore-like bodies, which were appressoria. Even after washing with water, the appressorium did not detach easily from the glass surface. He obtained the same result when the experiment was carried out on an apple surface instead of glass slide. He thus concluded that the spore must first come into contact with the host surface and adhere tightly in order to generate appressoria [43]. Emmett and Parbery expanded the definition and described appressorium as a significant penetration tool to break through the tough outer layer of plants, following adherence to the host surface [14].

Even though they were first observed in plant pathogens, appressoria are present in epiphytes, endophytes, saprobes, entomopathogens, and symbionts [4,14,41]. Appressoria are therefore not organs specifically evolved for basic infection processes of plant pathogens. Hence, for the rest of this review, we will maintain that appressoria are, “specialized cells or adhesion structures from which a penetration peg emerges that pierces the epidermal cell wall or enter the epidermis through stomatal apertures”.

## 3. Appressorial Categorization

Appressoria come in many different shapes and sizes and can be classified into two major groups (Figure 1): single-celled and compound appressoria [14]. Single-celled appressoria are the most common type among many species and occur at the end of germ tubes. This type of appressoria can be further classified into three groups: proto-, hyaline, and dark/pigmented appressoria. These hyaline and dark/pigmented appressoria have different shapes such as dome-shaped or lobed. Emmett and Parbery described proto-appressorium as little more than slightly swollen, hyaline germ-tube apices, adhering to the host surface with the formation of an infection-peg [14]. The proto-appressorium has a short germ tube, with no septum separating the apex from the germ tube. *Botrytis cinerea* elaborates a swollen cell at the apex of the germ tube in order to infect plants and was termed as a proto-appressorium [38]. *Venturia* (*Ven*.) and *Pythium* species too have formed ‘proto-appressoria’ when colonizing their hosts [14,52]. However, *Venturia inaequalis* has a melanin-deposited ring structure in the appressorial cell wall surrounding the penetration pore [27]. The second group of single-celled appressoria termed hyaline or non-pigmented appressoria are swollen and generally separated from the germ tube by a septum. Some genera in the orders of Erysiphales and Uredinales produce hyaline appressoria [14,52]. These hyaline appressoria are of different shapes including hook-shaped in *Bl. graminis* [53]; dome-shaped in *Oxydothis* species [54]; lobed in *Erysiphe* and *Neoerysiphe* species [55]; elongated, nipple-shaped in *Phyllactinia* species [56]; and sickle-shaped in *Phythium* species [57]. The third group is the dark or pigmented appressoria, usually thick-walled relative to the spores, either sessile or subtended by a germ tube, from which they can be separated by a septum. Take, for example, *Colletotrichum*, *Pyricularia*, *Polystigma*, and *Phyllosticta* species, which produce melanized appressoria. These melanized appressoria are also formed in different shapes such as dome-shaped in *Pyr. oryzae* [58] and lobed in *Col. boninense* [59]. The single-celled appressoria are formed not only terminally, such as on germ tubes, but can also be found laterally or intercalary, when formed on hyphae [14].

For all other multi-cellular appressoria, we grouped them into the compound appressoria category as in Emmett and Parbery [14] and Parbery and Emmett (1975) as cited in [14]. These structures are formed in various forms and include infection cushions, infection plaques, and expressoria [7,13,60,61]. Infection cushions are complex, multi-cellular appressoria with several penetration sites as in *Bot. cinerea* [13]; infection plaques are formed when runner hyphae differentiate to discrete clusters of parenchyma-like cells with multiple penetration sites as in *Psuedocercosporella herpotrichoides* [61]; expressoria are intrinsecus appressoria that are formed at the tip of hyphae of endophytes such as *Epichloe festucae* [7]. Most of the compound appressoria are hyaline [14]. Figure 2 exhibits different appressorial types produced by different fungal taxa.

## 4. Ultrastructural Studies of Appressoria

Knowledge gained from ultrastructural studies, combined with histological observations and physiological investigations on pathogenesis, enhances our understanding of host-fungal interactions at tissue and cellular levels. These studies provide information on how organelles and cytoskeletons reorganize during the appressorial formation and function. The information resulted from these studies is important as it provides notable targets for developing effective control measures for pathogens. For example, the application of biocontrol agents with fungicidal properties such as *Bacillus subtilis* on *Pyr. oryzae* inhibits spore germination, germ tube length, and appressorial formation, finally leading to reduced pathogenicity [64]. Appressorial formation has been studied at the ultrastructural level in *Col. truncatum*, *Col. lindemuthianum*, and *Pyr. oryzae* on sterilized dialysis membranes. To study appressoria at an ultrastructural level, the germlings were mainly analyzed by freeze-substitution with electron-microscopy [65]. Spore adhesion is an initial critical step in the establishment of fungal colonization, which is associated with hydrophobic interactions involving cell surface proteins [66]. This adhesion of *Col. lindemuthianum* may persist after the release of glycoprotein exudates at the substrate interface before germination [66]. As the spore germinates, the *Col. lindemuthianum* germ tube grows, and during its growth, the germling surface is surrounded by an extracellular matrix as visualized by colloidal gold [67]. The adhesion strength and its constituents differ among pathogenic fungi [68]. By using freeze-substitution, it was shown that non-germinated and germinated conidia of *Col. lindemuthianum* and *Col. truncatum* were coated with an electron-dense, finely fibrillar layer (Figure 3A) [67,69]. Negatively stained cells through a transmission electron microscope showed that fimbriae are protruded from the surface of germ tubes and appressoria as shown in Figure 3A [66].

The *Pyricularia oryzae* extracellular matrix is electron-dense and consists of amorphous and fibrous components [23], which is similar to the animal extracellular matrix [23,68]. Melanin is also one of the cell surface compounds of several plant pathogenic fungi including *Col. lindemuthianum*, *Col. truncatum*, and *Pyr. oryzae* (Figure 3B,C). Immunoelectron microscopy showed that electron-dense melanin layers were positioned between cell walls and plasma membranes of appressoria [23]. The electron-dense melanin layers are assumed to be involved in adhesion, turgor generation, cell wall rigidity, and protection against fungal enzymatic hydrolysis [23,67,69]. A study conducted on *Ven. inaequalis* disproved the importance of melanin for appressorial adhesion by demonstrating that appressorial adhesion to various surfaces is not affected by the melanin in appressorial cell wall [70]. Similarly, studies have shown that melanin is not essential for turgor generation but contributed to cell wall rigidity [71,72,73]. In addition, these functions are discussed in detail under Section 6. A study conducted on *Col. graminicola* demonstrated that melanized appressoria are protected against exogenous cell-wall-degrading enzymes whereas the non-melanized appressoria become lysed, suggesting that melanin has a primary role as a barrier for fungal cell-wall-degrading enzymes [34]. 

The conidial cytoplasm is comprised of ribosomes, mitochondria, lipid bodies, microbodies, vacuoles, multivesicular bodies, golgi bodies, woronin bodies, vesicles, and microtubules. When conidia are attached to substrates, a number of ultrastructural changes become evident [69]. In *Col. truncatum*, cristae of mitochondria became clearer because of the tremendously decreased electron density of the mitochondrial matrix. Microbodies also become prominent in germinating conidia, while their electron density increases [69]. During germination, the germ tube cytoplasm is often very dense, and the cytoplasmic components move from germinating conidia to the germ tube [69,74,75]. Eventually, the nucleus moves from the germinating cell to the germ tube to start mitosis. During mitosis, one of the resulting daughter nuclei moves into the germ tube apex, and the other one moves to the back of the germ tube [69,76]. The nuclear division is reported as a prerequisite for conidial germination and appressorial differentiation for both entomopathogens [77] as well as for plant pathogens [78]. Numerous microtubules concentrate in the conidial cell near the germ tube [75]. Vesicles are also abundant in the germinating spore and in the developing germ tube. Vesicles which condense near the germ tube apex are referred to as apical vesicles [65,74]. The migration and positioning of apical vesicles of germling apex are influenced by the interaction of vesicles and cytoskeletons [74,79].

Once the germ tube is attached to the substrate, it starts to differentiate into an appressorium [75]. Early in the appressorium development, the distribution of microtubules changes from a parallel direction with the original longitudinal axis of the germling to the parallel region of a cell exhibiting lateral growth [74,80]. The vesicles originally sent towards the apex change their distributions around the cells and cause cellular ballooning [80]. During appressorium formation, the germ tube cytoplasm migrates into the swollen apex [74,76]. Eventually, a septum develops to separate the mature appressorium and the germ tube [69,74,75,76].

Near the center of the fully developed appressorium (Figure 3D) which directly contacts with the substrate, the cell wall becomes extremely thin. Apical vesicles along with multivesicular bodies, filasomes, and a few microtubules become concentrated in the apex of the appressorium [65,79]. Eventually, a small penetration peg grows from the thin-walled region. As evident in *Col. truncatum* [69] and *Pyr. oryzae* [79], many small apical vesicles are present in the apex of the penetration peg.

## 5. Appressorial Infection Process

In this section, we will review our current understanding of appressoria and their common mechanisms in pathogenesis related to selected fungal taxa. Pathogenesis involves a series of events that eventually can lead to disease development in a host. Here, we examine the main stages in the disease cycle of fungal pathogens on plant hosts (Figure 4), beginning with the landing of spores, their attachment, and entry into the plant tissues via appressoria formation and penetration. Appressoria are produced when fungi need to access nutrients from their hosts [60]. The very first event in a disease cycle is the contact of the fungus with a susceptible host plant [81].

When the microclimatic conditions are favorable and the nutrients are available, spores or the primary inocula will start germinating [81]. Resting spores may wait until the arrival of new roots in their vicinity, or else motile zoospores may be attracted to their hosts. For example, *Phytophthora cinnamomi* is attracted to *Persea americana*, and *Ph. citrophthora* is attracted to citrus plants [81].

Spreading spores land on aerial plant surfaces via wind, rain splash, or facilitation by some animal activities. Some of these spores initiate the germination process immediately, when coming in contact with water or under humid conditions [81]. Some spores require a host stimulus to initiate their germination. Spores of many facultative pathogenic fungi germinate more rapidly in the presence of certain available nutrients. After landing on the host plant surface, spore attachment, germ tube formation, and entry into the host may follow to establish infection [72]. Fungal entry into the host occurs via natural openings, via wounds, or through intact surfaces [81]. Most of the obligate pathogens enter via stomata, and many facultative fungal pathogens enter directly by penetrating through intact plant surfaces [60].

Upon recognition of physical clues on the host plant surface, such as surface hardness and hydrophobicity, spores germinate and form germ tubes. Then, potentially different appressoria are formed on the plant surface, depending on the location of the fungal entrance mentioned above (sometimes over the openings) [81]. These appressoria vary from simple swollen cells to complex infection cushions (Figure 5), which anchor the fungus before and during penetration [81]. Penetration pegs (a thin hypha) usually originate from the center of the appressorium and extend towards underlying tissues assisted by cell-wall-degrading enzymes and through physical force [71,81]. After penetration through the outer cuticle, fungal growth and nutrient utilization vary significantly, depending on the nature of the host-fungus interaction [81,82]. Different fungal species acquire nutrients from their host plants in different ways, and their effects on the hosts differ accordingly.

## 6. Molecular Mechanisms Involved in Appressorial Infection and Evasion of Host Defenses

Host-fungal interactions include pathogen’s offense, host defense, and pathogen’s counter-attack. In each interaction, if the host plant can ward off the invading fungus, disease resistance develops, but if the fungus can overcome host defense, then disease develops. In the cases of endophytic and saprobic fungi, successful colonization occurs when the fungus establishes a foothold on host plants. During their coevolution with plants, fungi have developed counterproductive mechanisms to elude the recognition by plant receptors or disturb the signaling cascades of the plant’s innate defenses [5,28].

The adhesion of fungal spores to the surface of a plant initiates disease establishment. When a fungal spore lands on the cuticle or the surface wax layer, the fungi may sense the molecules that inhibit its entrance into the plant. Upon receiving these topographic and hydrophobic signals, fungal spores release adhesives specific to different fungi [28]. The composition of these adhesives differs among different groups of fungi, depending on the plant surface and environmental signals. The compositions of these adhesives include glycoproteins (*Stagonospora nodorum* (*St*.) [85]), hydrophobins [28,86,87], chitosan and chitin-binding proteins [26,28], lipids, and polysaccharides [2]. Many studies have shown the roles of these adhesives in attaching appressoria to plant surfaces [28]. In addition to their role in adhesion, molecules such as hydrophobins shield fungal spores from being recognized by plant immune cells, as well as from being recognized by the insect immune systems (*Aspergillus fumigatus* [86]). Furthermore, they contribute to the surface hydrophobicity of aerial hyphae to protect the aerial hyphae against waterlogging (e.g., in *Pyr. oryzae* [87]). The importance of chitosan-synthesizing enzymes in the germling adhesion was demonstrated in *Pyr. oryzae* [26]. However, the function of these adhesives is not conserved in filamentous fungi. They vary depending on their infection strategies and the mechanisms they employ to invade a host. Secretion of these adhesives is a passive process for some fungi such as *Pyr. grisea* [60] which allows for rapid attachment. For other fungal pathogens such as *Col. graminicola*, energy is required to complete the process. For example, it takes more than 30 min for *Col. graminicola* to complete the attachment process [88]. Similarly, in entomopathogens such as *Metarhizium anisopliae*, the insect cuticle influences the conidial adhesion, germination, and appressorial differentiation [89]. Entomopathogenic conidia most often attach to the bases of hair sockets or on the intersegmental membranes, where cuticle flexibility allows the conidial attachment [90]. Similar to plant pathogens, these entomopathogenic fungi also secrete an adhesive mucus via conidia prior to germination [90]. Upon germination, *M. anisopliae* produces a short germ tube at the tip or the middle region of the conidia, which then proceeds to develop the appressorium on the epicuticle. These appressoria vary in morphology such as cupped, clavate, curly, and globose-shaped structures [90].

In addition to adhesives, spores and various fungal structures, including the infection hyphae, appressoria, and haustoria, cells are often surrounded by an extracellular matrix (ECM). This ECM is implemented during the fungal adhesion of many groups of fungi [91]. However, the composition and the structure of ECM can differ among different groups of fungi, depending on the surface and environmental signals; well-studied examples include *Bipolaris* species [92] and *Cochliobolus* species [37]. These ECMs consist of glycoproteins as in *St. nodorum* [85] and lytic enzymes such as esterases and cutinases as in *Col. graminicola* [93]. Furthermore, the presence of these enzymes in the ECMs facilitates breakdown of the cuticle, which allows the absorption of nutrients required for spore germination as seen in *Bl. graminis* [2]. In addition to the chemical responses stated above, some fungi including *Pyr. grisea*, *Col. graminicola*, and *Uromyces appendiculatus* adhere physically to the hydrophobic substrate upon recognizing the barriers [2].

After successful attachment, a series of developmental events occur, including spore germination and germ tube elongation in response to plant and fungal signals [94]. Different plants secrete different signals, and such signals are perceived differently among different fungi. Plant signals such as nutritional availability [48] and hydration as for *Pyr. grisea* [2] stimulate spore germination. Chitin-binding proteins that are secreted by the fungus mediate the perception of environmental cues such as hydrophobic surfaces to facilitate spore adhesion and induce germ tube development as demonstrated in *Pyr. oryzae* [17]. After formation of a germ tube, it acts as the main site for plant-signal perception [2]. During the germ-tube-elongation process, it receives plant signals such as surface hydrophobicity and hardness [17], topographic signals [18], cutin monomers [19], and ethylene signals [2]. On receipt of these signals, germ tubes differentiate and form appressoria [48]. This series of events determine the success of infection. Stimulation required for spore germination differs according to the nutritional mode of the fungus. For example, in rust fungi, recognition of the physical surface signals, such as ridges and the lips of stomatal guard cells, induces appressorium differentiation [2]. Furthermore, past studies have shown the role of calmodulin-dependent protein kinases in transducing surface signals and Ca2+ ions, which trigger appressorium differentiation from the germ tube not only in plant pathogens such as *Uromyces appendiculatus* [95], but also in entomopathogens such as *Zoophthora radicans* [96]. Hydrophobins secreted by fungi facilitate spore adhesion, specifically class I hydrophobins involved in attaching the germ tube apex to the plant surface, inducing cellular differentiation into an appressorium [87]. Some of the chemicals secreted by plants during innate defenses are recognized by the fungus as inducers for spore adhesion, germination, and appressorium formation. These inducers include fatty acids, fatty alcohols, flavonoids, and aldehydes [97]. Fatty alcohols released during the degradation of plant cutin of avocado hosts were shown to be the main inducer of appressorium formation in *Colletotrichum* species [98]. Furthermore, alcohols and aldehydes comprised in the cuticular wax promoted germ tube gemination, differentiation, and appressorial formation in *Pyr. oryzae* [30] and *Bl. graminis* [99], respectively. Similarly, plant flavonoids produced by legume roots demonstrated their ability to induce spore germination in *Nectria haematococca* [100]. While fungal appressoria differentiate, the fungi are required to evade host defenses simultaneously. A previous study identified a host-selective HC-toxin in *Cochliobolus carbonum* that is associated with appressorium differentiation. This toxin has been confirmed to inhibit the histone deacetylase activity. Histone deacetylase is involved with the defense of host plants and functions to prevent the activation of defense genes [101].

All the mechanisms from spore adhesion to the appressorial differentiation and host invasion are dependent on the recognition and transduction of host or environmental signals to its downstream by signal transduction pathways, such as cyclic AMP, G proteins, and mitogen-activated protein (MAP) kinase pathways [28,30,33,102,103]. For example, appressorial formation in *Pyr. oryzae* is induced by a series of cAMP/PKA and DAG/PKC signaling cascades triggered due to the presence of cutin monomers, resulting from the cutinase-mediated degradation of the plant cuticle [30]. The highly conserved MAPK signaling pathway plays a crucial role in appressorium formation and penetration. Many MAPK proteins (Pmk1, MEKK-Mst11, Osm1 in *Pyr. grisea*) have been shown to play crucial roles in forming appressoria [28,33]. For example, the targeted deletion of these MAPK genes such as *Pmk1* results in non-pathogenic mutants unable to produce appressoria [102]. Targeted deletion of MAPK genes, *Bbhog1* in *Be. bassiana*, resulted in reduced pathogenicity, demonstrating their importance for spore viability, adhesion to the insect cuticle, and appressorium formation in entomopathogens [104]. Furthermore, *Bbmpk1* in *Be. bassiana* is another gene characterized to be involved in fungal adhesion to the cuticle, appressorium formation, and fungal penetration [105]. Homologs of this gene have been shown to be involved in similar functions in other plant pathogens such as *Pyr. grisea* [106], *Col. lagenarium* [21]. In addition to the MAPK signaling pathway, both cAMP and calcium signaling cascades regulate appressoria development and pathogenicity in plant pathogens such as *Pyr. oryzae* [33] and *Colletotrichum* species [103], and entomopathogens such as *M. anisopliae* [107]. A major cAMP-dependent protein kinase A (PKA) is related to the production of functional appressoria (encoded by the *cpkA* gene). PKA regulates the mobilization of lipids and carbohydrates to the appressorium. This was confirmed by *cpkA* mutation studies in *Pyr. grisea* [72], *Col. trifolii* [108], *Bl. graminis* [22], and *M. anisopliae* [107]. PKA mutations in these organisms resulted in appressoria without the capacity to penetrate cuticles and cause disease [22,72,107,108]. Based on current knowledge, many signaling cascade genes are involved in both appressorium formation and subsequent processes during pathogenesis in many phytopathogenic fungi [28].

These intracellular signaling cascades regulate the activity of transcription factors that are critical for activating gene expression in response to extracellular signals. Gene characterization research provides ample evidence for transcription factors regulating changes to gene expression during appressorium development [28,36,39,109]. These studies demonstrate the diverse roles played by these transcription factors during appressorium morphogenesis and pathogenicity of *Pyr. oryzae*, *Col. orbiculare*, *Sc. sclerotium*, and many other pathogens [28,36,39,109,110,111,112,113]. The transcription factors GcStuA in *Glomerella cingulata* [109], and Mstu1 in *Pyr. oryzae* [110], regulate conidial reserves during the appressorial turgor generation; Moswi6 and MoMst12 in *Pyr. oryzae* are required for the penetration peg formation [39]; Hox7 in *Pyr. oryzae* is vital for appressorial development [36]; SsNsd1 and Ssams2 in *Sc. sclerotium* are responsible for the infection cushion formation and differentiation [111,112]; and CoHox3 in *Col. orbiculare* is responsible for appressorial formation and maturation [113].

Appressoria facilitate fungal penetration into host tissues; however, the mechanism behind this process has been controversial and subjected to comprehensive studies [2,8,34,71,72,73,114]. During appressorium maturation, fungi synthesize and accumulate both glycerol and melanin. They then form a thick melanin layer on the inner appressorial wall, as a response to combat the plant’s innate defenses and induced immunity. During appressorium development and maturation, various cellular pathways contribute to the accumulation of glycerol in the appressorium by the use of dihydroxyacetone phosphate, dihydroxyacetone, glyceraldehyde, or triacylglycerol as precursors [72]. The accumulated glycerol draws water into the cell through osmosis and facilitates the generation of hydrostatic turgor pressure [2], and the appressorial wall remains in contact with the plant cuticle. In pathogens such as *Pyr. grisea* and *Col. lagenarium*, a series of reactions involving polyketide synthesis occur, leading to the polymerization of 1, 8-dihydroxynaphthalene (DHN). This results in the production of DHN-melanin [8]. A study conducted on *Col. graminicola* showed that melanin is not required for solute accumulation and turgor generation, contrary to that seen in *Pyr. oryzae*, *Pyr. grisea*, and *Col. lagenarium* [34]. This melanin that is deposited between the appressorial cell wall and the plasma membrane binds to the plant surface to lower the porosity of the appressorial wall, blocking osmolytes [8,71]. As a result of glycerol accumulation and cell wall melanization, an enormous hydrostatic turgor pressure is generated [8,71]. This extreme pressure is translated into a physical force that provides the structural rigidity required for the penetration hypha to penetrate through the cuticle and epidermal layers of the host forcibly [71,73]. Bastmeyer et al. [73] visualized and quantified the exerted physical force of the penetration pegs on an optical waveguide. Bechinger et al. [114] showed that the exerted force by matured appressoria increases rapidly, suggesting that the exerted physical force is not due to the sudden release of turgor but sustained application of the physical force by the penetration peg. In entomopathogens, appressoria produce a narrow peg-like structure, which penetrates the epicuticle and reaches the endocuticle, where they differentiate into a lateral penetrant structure. These structures lyse the endocuticle layer using cuticle degrading enzymes [115] to enter into the host [90]. Unlike most plant pathogens, host penetration of *M. anisopliae* is mainly due to the chemical dissolution of structures rather than the mechanical force [90].

Appressorial pore is the point of infection which facilitates the contact of the fungal plasma membrane with the plant surface. At the appressorial pore, actin cytoskeletons organize as a network at the base of the appressorium, scaffolded by septin [116,117]. Septin are small morphogenetic guanosine triphosphates (GTPases) that assemble as a large ring surrounding the F-actin network at the appressorium pore to provide the cortical rigidity [116]. These septin also act as diffusion barriers to confine domain proteins that function in generating the membrane curvature and protrusion of the penetration peg, which is required to rupture the host cuticle [116]. In addition, studies have shown that the catalyzation of reactive oxygen species burst by NADPH oxidases (Nox2) are required for the appressorium repolarization process [117]. This Nox2-NoxR complex is essential to organize the septin ring and F-actin network at the appressorium pore [79,117,118]. Gene mutation studies involving septin and F-actin have resulted in the failure to differentiate appressorial pores, as well as stunted penetration pegs that failed to elongate and break the cuticle [79,116,117,118].

During penetration of the cuticle into the plant cell, fungi must overcome the barrier of the plant cell wall, which is composed of many polymers such as cellulose, xylan, and pectin [5]. For fungi which produce non-melanized appressoria or inconspicuous appressoria such as *Bot. cinerea*, it is assumed that cell-wall-degrading enzymes play a more prominent role during penetration than the physical force [106]. Gene-expression-profiling studies have shown that cell-wall-degrading enzymes secreted through the penetration peg perform enzymatic softening of the substratum and allow fungal hyphae to penetrate plant cells. Fungi with highly melanized appressoria such as *Col. gloeosporioides* show evidence of cell-wall-degrading-enzyme secretion during penetration [119]. Fungi with weakly melanized appressoria and melanin-deficient mutants such as *Cochliobolus* species exhibit a lot higher cell-wall-degrading-enzyme activity during the penetration of hyphae [1,5,120,121,122,123,124]. During penetration, cellulases, cutinases, endo- and exo-polygalacturonases, pectinases, pectate lyases, polymethylgalacturonases, pectic methylesterases, and rhamnogalacturonases act in plant pectin degradation. Examples include four functional pectate lyases in *Nectria haematococca*, five endo-PGs in *Bot. cinerea*, and four xylanases in *Coc. carbonum* and *Pyr. grisea* [5,120,121,122,123,124]. Targeted silencing of single genes is difficult due to the redundancy of the cell-wall-degrading genes responsible for the plant cell wall degradation, hence identifying their exact role in host penetration is challenging [125]. However, Tonukari et al. [121] demonstrated that the inactivation of *SNF1* in *Coc. carbonum* resulted in reduced penetration of the fungus. Similarly, reduced or impaired penetration was observed in the knockout mutants of xylanases [122] and cellulases [123] in *Pyr. oryzae*. *Pyricularia grisea* cutinase2 genes are required for the appressorial differentiation and host penetration [124]. In all these studies, the host penetration is reduced or impaired as a result of gene mutation, suggesting that the physical force exerted by the penetration peg alone is not sufficient, and enzymatic degradation of the cell wall also assists in the host penetration in some fungi. Similar to plant pathogens, entomopathogens also secrete toxins and degrading enzymes to reduce the host defense, transporters which provide protection against host defenses or components of signal transduction pathways that are necessary to sense the host environment [41,105,107,126,127,128].

When a fungus initiates plant infection, pathogenic fungi and their proteins are recognized by plant pattern recognition receptors and induce pathogen-associated molecular patterns-triggered immunity responses. As a response, fungal taxa deploy a repertoire of effectors to suppress the plant immunity through a penetration peg [24,129,130]. Several studies have shown evidence of specialized focal secretion of effectors at the penetration peg and an extending primary infection hyphae in *Col. higginsianum* [25], *Col. orbiculare* [24], and *Pyr. oryzae* [129]. Effector genes such as *pep1* and *pit2* have been extensively analyzed in the corn smut pathogen *U. maydis* [42]. Gene deletion studies on these effectors have proven the successful induction of plant defense responses [42,131]. It was also revealed that some effectors have the ability to target plants’ secondary metabolite pathways, preventing plants’ induced immune responses, such as lignification [131]. Unlike the biotrophic fungal effectors, necrotrophic fungal effectors such as polyketide toxins, nonribosomal peptide toxins, necrosis- and ethylene-inducing peptide 1 (Nep1), and Nep1-like proteins can induce cell death [5]. Many of these effectors are well studied and have shown to facilitate fungal colonization by compromising the host defenses and contributing to the establishment of symbiotic and pathogenic relationships [5]. However, plants have evolved mechanisms that recognize these effectors using R proteins and induce effector-triggered immune responses. As a response to these effector-triggered immune responses, fungi secrete effectors encoded by avirulence genes, which target the R proteins [130]. Such effectors have been identified in many hemibiotrophs, such as *Pyr. oryzae* [132], *Col. higginsianum* [31], *Verticillium dahliae*, and *F. oxysporum* [133].

Appressorium pores and penetration pegs act as key hubs for the signaling and secretion of host immune suppressors during plant infections. Therefore, fungal appressoria have evolved mechanisms to evade plant defense responses from the initial spore attachment to when they completely colonizes the plant tissue, as discussed above.

## 7. Genomics, Proteomics, and Metabolomics of Appressoria

Since the unraveling of the first genome for *Pyr. oryzae* [134], many genomes have been published for appressoria-producing fungi, which include plant pathogens, entomo-pathogens, saprobes, and endophytes [31,135]. As new genome data emerge for different groups of fungi, DNA sequences and probes become important tools for phylogenetic analyses related to fungal identifications, and for the detection and manipulation of the expression of genes involved in pathogenesis. Many pathogenicity-related genes have been identified and characterized over the years, such as *MaSte12* in *M. acridum* [35], transcriptional regulators in *Pyr. oryzae* [36], *CBP1* [17] and *MPG1* [87] in *Pyr. grisea*, *SsNsd1* in *Sc. sclerotiorum* [29], and effector-encoding *Pep1* in *U. maydis* [42]. Appressorium formation and function are complex morphogenetic processes that are tightly linked to genetic regulation. They are intricately coordinated by an array of genes and signaling pathways. Many such genes involved with appressorial formation and function have been identified and characterized, including exocyst components and proteins in *Pyr. oryzae* necessary for appressorial repolarization and host cell invasion [79], *ATG8* in *Pyr. oryzae* related to appressorium maturation and infection [136], *MoMps1* related to appressorial function in *Pyr. oryzae* [39], *MAC1* in *Pyr. grisea* [137], *MoSfl1* in *Pyr. oryzae* [33] and *MaSte12* in *M. acridum* [35] involved in appressorium formation, *BcPIs1* related to appressorial penetration in *Bot. cinerea* [38], *CgPKS1* involved with appressorial melanization in *Col. graminicola* [34], *SsNsd1* responsible for infection cushion formation in *Sc. sclerotiorum* [29], and *MAF1* and *CBP1* involved in appressorial differentiation in *Col. lagenarium* [21] and *Py. grisea*, respectively [17]. Regulation of these genes is needed to ensure a successful disease establishment, in the case of pathogens or successful host colonization, and in cases of endophytes and saprobes [2,28,138].

The most comprehensively studied appressoria are the highly melanized ones of *Pyr. grisea* [17,87], *Pyr. oryzae* [33,39], *Col. higginsianum*, and *Col. graminicola* [31]. A second well-studied group includes the non- or lightly melanized appressoria of *Bl. graminis* and other oomycetes [138], and the appressoria of many entomopathogenic fungi [35,41]. A plethora of infection-related gene families involved in spore adhesion and germination, appressorial formation, and host penetration have been revealed through comparative genomic and transcriptomic studies in many fungi. These gene families include ATP-binding cassette (ABC) type transporter, major facilitator superfamily and ion transporter families involved in cellular transportation, cholesterol, phospholipid, phosphatidylcholine and peroxisome biosynthesis gene families involved in lipid and fatty acid metabolism, carbohydrate metabolism gene families, proteases, protein ligases and glutamate dehydrogenase gene families related to protein and amino acid metabolism, secondary metabolism gene families encoding melanin biosynthesis, polyketide synthesis, cytochrome P450s, non-ribosomal peptide synthetases (NRPS), chitinases, glucanases, glucosidases, polysaccharide dehydrogenases, cutinases, lignin peroxidase gene families involved in cell wall degradation, gene families encoding transcription factors, pathogenicity-related gene families involved in GAS homologs, hydrophobin and candidate secreted effector protein (CSEP) synthesis, and gene families related to signal transduction such as MAPK, phospholipases, adenylyl cyclases, and protein kinases [28,31,32,40,41,133,134,135,139,140]. Previous studies report that these gene families are involved in infection structure formation, including producing secreted effectors, adhesives, pectin-degrading enzymes, chitin-degrading enzymes, secondary metabolic enzymes, transporters, proteases, and peptidases, with many of these gene families expanded in pathogenic fungal genomes. For example, most of the pathogenicity-related gene families encoding secreted effectors, pectin-degrading enzymes, secondary metabolism enzymes, transporters, and peptidases are expanded in *Col. higginsianum* [31]. These expanded gene families principally help in the increased production of their encoding proteins and also provide the material for specific adaptations leading to the evolution of new functional systems [31,41,126,139,141]. For instance, a comparative analysis identified expansions in cutinase, cytochrome P450, and serine protease gene families in *Pyr. grisea* compared to non-pathogens, and suggested that these expansions might be associated with the pathogenic evolution of *Pyr. grisea* [134]. Encoded proteins by these gene families serve as the raw materials for behavioral and physiological adaptations to overcome host defenses. For example, expansions of cellulose-binding module (CBM) genes in *Sc. sclerotiorum* and *Bot. cinerea* demonstrate their stronger preference for vegetative plant tissues, confirming that these gene family expansions are involved in adaptations to specific ecological niches [135].

Furthermore, the transcriptome analyses of these fungi during spore germination and appressorium formation have helped reveal the genes with significant changes in their expression. In the *Pyr. oryzae* genome, approximately 21% of the genes (2,154 genes) showed differential expressions by more than two-fold during infection. The majority of these genes exhibited increased gene expression during spore germination and appressorium formation [40]. Specifically, 357 genes were differentially expressed during appressorium formation. Among them, 240 genes increased their expression, whereas 117 reduced their gene expression. The genes that showed a significant increase in expression when the genes were involved in protein and amino acid degradation, lipid metabolism, secondary metabolite synthesis (melanin biosynthesis), and cellular transportation. In contrast, the genes with significantly decreased expression during appressorium induction were those that were involved in protein synthesis related to ribosome biogenesis. Transcriptome analysis showed differential expression of *Mgd1*, *GAS1*, and *GAS2* genes that are related to appressorial formation, and *SPM1* involved with host penetration. The functional characterization and differential expression of these genes solely or in combination with other genes establish their involvement in appressorium morphogenesis and provide an understanding of protein degradation during appressorium functioning [40]. In a similar study conducted on transcriptional profiling of *Pyr. oryzae*, different genes upregulated at different stages of appressorial formation and function were identified [32]. During spore germination and early appressorium development, several genes were significantly upregulated, such as *cdc14*, *Mis14*, *sudA*, and *tinA*, all four genes under the transcriptional control of the Pmk1 MAP-kinase pathway. According to the transcriptional profile, genes regulating acetyl-CoA synthesis responsible for appressorium morphogenesis, several polyketide synthase genes involved in the melanin biosynthesis pathway, genes involved in the beta-oxidation of fatty acids, and several genes encoding cell-wall-degrading enzymes (cutinases, xylanases, polygalacturonases, cellulases) showed increased expression at appressorial development and maturation stages. Furthermore, several transporter genes such as sugar transporters, MFS transporters and multidrug and toxin extrusion (MATE) family of transporters, and ABC transporters upregulated during appressorium function, and the latter one also during host penetration [32]. Another study identified increased expression of *mas1*/*mas2* genes responsible for germination and appressorium formation, genes encoding tetraspanin involved in appressorium function, and genes involved in MAP kinase and cAMP-dependent signal transduction pathways in *Sc. sclerotiorum* and *Bot. cinerea* [135]. In addition to these plant pathogens, entomopathogenic *M*. *acridum* also demonstrated the importance of tetraspanin-encoding *MaPls1* for conidial germination, appressorium formation, and turgor pressure generation [127], and perilipin-encoding *Mpl1* responsible for the turgor generation in *M. anisopliae* [128]. Similar to previous studies, O’Connell et al. [31] identified five gene categories (encoding transcription factors, secondary metabolism enzymes, CSEPs, CAZymes, and transporters) with increased expression patterns relevant to pathogenicity in *Col. higginsianum* and *Col. graminicola* at the appressorial formation stage. For example, genes encoding CAZymes essential for the host penetration such as cutinases, cellulases, hemicellulases, and pectinases were identified. Prior to the penetration and during biotrophy stages, a significant increase in gene expression was observed for 12 different secondary metabolism gene clusters, including CSEP-encoding genes (*ChEC6* being the most highly induced) for host manipulation. Similar genomic and transcriptomic studies conducted on different plant and entomopathogenic fungi such as *Be. bassiana*, *Bl. graminis*, *Candida albicans*, *Cordyceps militaris* (*Cor.*), *F. graminearum*, *M. anisopliae*, *M. acridum*, *Pyr. grisea*, and *Uromyces fabae* identified similar genes homologous to the above-mentioned gene categories involved in surface attachment, appressorium formation, and host penetration. These discoveries lay the foundation for future research and design of novel strategies for disease control such as improving biocontrol efficacies of fungicides and fungal insecticides [21,33,34,35,38,39,40,41,137].

Comparative proteomic studies facilitated the identification of proteins involved in the formation of functional appressorium [29,141,142,143]. Franck et al. compared the proteome of an appressorium-producing *Py. oryzae* with a mutant strain that produces a non-functional appressorium to determine the proteins essential for the appressorium function [141]. According to Franck et al. [141], 193 proteins specific to germination and appressorium formation were identified in *Pyr. oryzae*. The same study discovered 59 proteins involved in the cAMP signaling pathway during appressorium formation. From the total proteins identified, 193 proteins (6% of the total proteins) were identified to be involved in conidial germination and appressorial formation. Among them, 40 were specific to appressorium formation. A comparison of the appressoria-enriched cDNA library of the rust pathogen *Phakopsora pachyrhizi* (*Pha.*) with existing expressed sequence tags (ESTs) revealed 29 ESTs specific to the appressoria-enriched library. Among these, 35% of the ESTs are important for autophagy, mitosis, and fungal metabolism (cholesterol biosynthesis and amino acid metabolism) involved in appressoria formation. In addition to those, serine/threonine protein kinases, P-type ATPases involved in appressorial morphogenesis, subtilase-type proteinases involved in the appressorial development, isocitrate lyases related to lipid metabolism (synthesis of glycerol), and septin involved in urediniospore germination and germ tube elongation were also identified to be in abundance [143]. Similar proteins were identified in *Aspergillus oryzae*, *Be. bassiana*, *Bot. cinerea*, *Cor. militaris*, *F. oxysporum*, *Glomus intraradices*, *M. roberstii*, *Ophiocordyceps sinensis*, *Ph. infestans*, *Pyr. grisea*, and *Sc. sclerotiorum* [29,41,141,142,143,144]. In almost all of the studies, a higher expression is evident in proteins responsible for the degradation and recycling of the proteins, and cell contents. This can be explained as the fungus-need energy to carry out functions such as spore germination, germ tube elongation, and appressoria formation prior to host penetration. Hence, this is achieved by breaking down spore content. Furthermore, the proteins related to lipid, glycogen, and sugar metabolism are necessary for the synthesis of glycerol in the appressorium which is considered as one of the factors contributing to the turgor pressure during host penetration. The significance of the proteins involved in the breakdown and recycling of cellular components has been demonstrated in *Pyr. oryzae*, *Pyr. grisea* [136], and *Col. orbiculare* [122].

In another study, Liu et al. [138] performed an untargeted metabolomic analysis to profile the metabolome of developing appressoria and identified significant changes in six key metabolic pathways in *Pyr. oryzae*. According to the KEGG analysis, these pathways include degradation of lipids, degradation of carbohydrates, arginine synthesis, sphingolipid synthesis, sterol synthesis, and phospholipid metabolism. As one of the early intermediates of the sphingolipid biosynthesis pathway, ceramides were shown to be essential for normal appressorial development, specifically for appressorial morphology and penetration peg formation. It was believed to affect the anaphase of mitosis and degradation of nuclei in conidia. They are shown to be accumulated during appressorial morphogenesis and metabolized during maturation in *Pyr. oryzae*. Similar processes were identified in the Asian soybean rust fungus, *Pha. pachyrhizi* [145].

Genomics, transcriptomics, proteomics, metabolomics, and comparative approaches facilitated the identification and functional characterization of differentially expressed genes, and revealed their functions related to appressorial formation and function. For example, using the targeted gene deletion approach, several studies have revealed the importance of hydrophobin genes (*sc3*, *sc1*, and *sc4* in *Schizophyllum commune*; *mpg1* in *Pyr*. *grisea*; *cu* in *Ophiostoma quercus*; and *crp* in *Cryphonectria parasitica*) for spore attachment, germ tube attachment, and blockage of plant recognition in different fungi [28,86,87]. Most importantly, major genes involved in the signaling pathways that regulate the appressorial development processes (*mac1*, *sum1-99*, *pmk1*, and *cpkA* in *Pyr. grisea*; *chk1* in *Coc. heterostrophus*; *MAF1* in *Col. lagenarium*; *bmp1* in *Bot. cinerea*; *fmk1* in *F. oxysporum*; *ptk1* in *Pyrenophora teres*; *pka1* in *Sc. sclerotiorum* and *M. anisopliae*; and many more) have been characterized in many fungi [2,20,21,28,102,103,107]. For example, the importance of these signal pathways is evident when a gene *mac1*, encoding a cAMP signal is deleted in *Pyr. grisea*. This resulted in the complete loss of appressorium formation in the mutant [137]. Furthermore, studies on the expression of genes, which control the turgor development in the appressoria, have shown functionally the importance of each of the genes [2,106]. Similarly, the function of cell-wall-degrading enzymes in the penetration process [5] and effectors and *Avr* genes for the suppression of host defense have been reviewed and demonstrated in many studies [2,24,25,42,130,131,132].

With improvements in genomics and biotechnological techniques, further gene functions related to appressorium formation and function will be revealed. Specifically, gene silencing and modification technologies will facilitate discovery of new genes and the key signaling molecules important for plant-fungal interactions. An in-depth understanding of these processes will provide an unprecedented opportunity for plant pathology research and to the improvement of agriculture practices.

## 8. Conclusions

While most of the studies discuss appressoria formation as an independent event in the infection process that exclusively occurs in pathogenic taxa, we discuss it as a part of a sequence of events that occurs when a fungus colonizes a host. In this study, we provide a comprehensive review of our current understanding of the diversity of appressoria in fungal taxa. We described how appressoria are classified among fungal taxa and the changes that occur in their ultrastructure during the host attachment and colonization. The application of functional genomics techniques reveals signaling events between the fungus and the host. Furthermore, these ‘omics’ studies identify genes encoding the function and formation of different morphogenesis stages in various fungi such as *Col. graminearum*, *Col. gloeosporioides*, *Pyr. grisea*, and *Bot. cinerea*. This knowledge base is likely to grow due to the exponential growth of genomic and transcriptomic studies, and the discoveries from these disciplines are discussed together with the potential for further developments. The applications of transcriptomics, proteomics, secretomics, and metabolomics offer significant opportunities to advance the understanding of appressorium formation and function during host-fungal interactions. Combining these approaches in a single study has many advantages over a single ‘omics’ study based on a single technique alone. Knowledge of genetic mechanisms and appressorial morphogenesis is important for designing control strategies against plant fungal diseases. An example of using this knowledge is the use of anti-penetrant fungicides to control *Pyr. grisea*-related fungal disease. The fungicide interferes with the melanin biosynthesis of the fungus, hence, producing non-functional or less effective appressoria [48].

## Figures and Tables

**Figure 1 pathogens-10-00746-f001:**
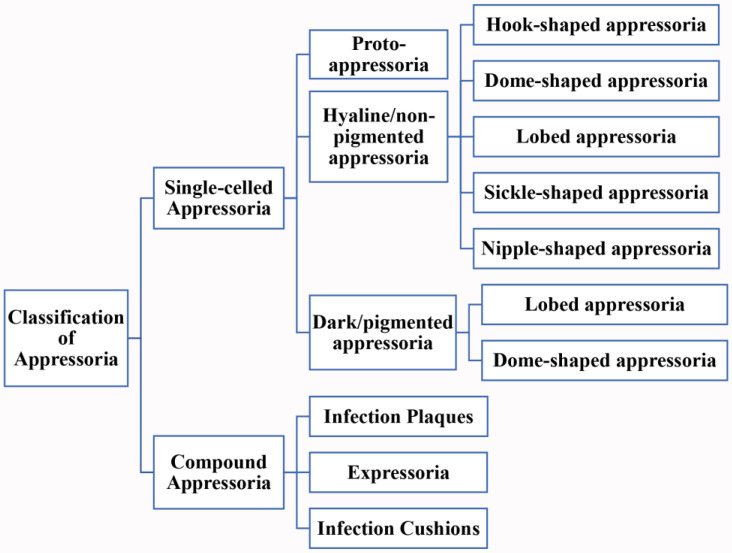
Classification of appressoria.

**Figure 2 pathogens-10-00746-f002:**
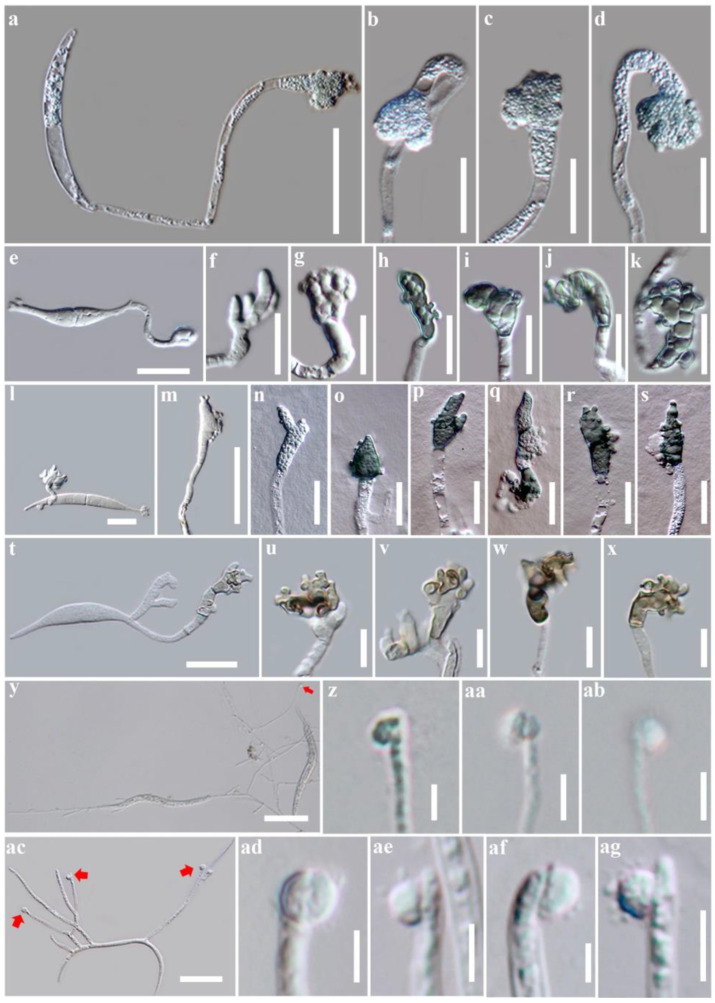
Single-celled appressorial types produced by fungal taxa. (**a**–**d**) Single-celled, hyaline, irregular-shaped appressoria in *Oxydothis garethjonesii* [54]; (**e**–**k**) Single-celled, light green, irregular-shaped appressoria in *O. metroxylonicola* [54]; (**l**–**s**) Single-celled, hyaline to dark green, irregular-shaped appressoria in *O. metroxylonis* [54]; (**t**–**x**) Single-celled, hyaline to brown, irregular-shaped appressoria in *O. palmicola* [54]; (**y**–**ab**) Single-celled, hyaline, globose to subglobose appressoria in *Leptosporella arengae* [62]; (**ac**–**ag**) Single-celled, hyaline, globose to subglobose appressoria in *Neolinocarpon rachidis* [63]. Scale bar: (**a**,**y**,**ac**) = 50 µm; (**e**,**t**) = 20 µm; (**b**–**d**,**f**–**s**,**u**–**x**) = 10 µm; (**z**–**ab**,**ad**–**ag**) = 5 µm.

**Figure 3 pathogens-10-00746-f003:**
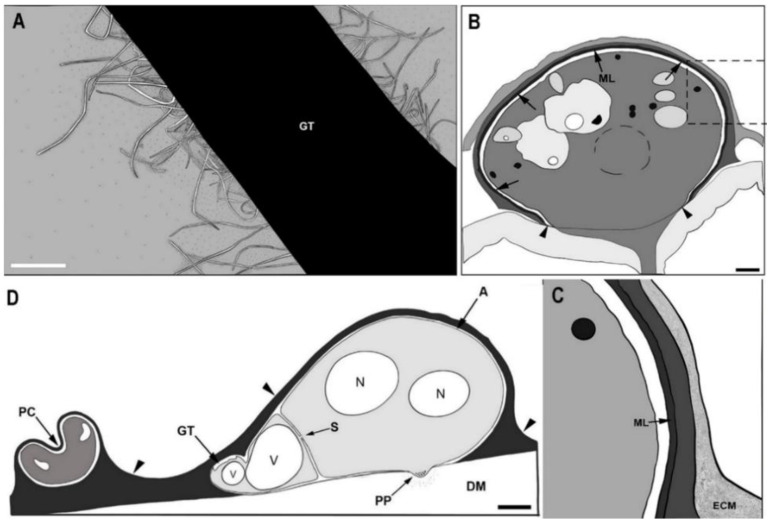
Ultrastructures of appressoria and extracellular matrix. (**A**) Ultrastructure of extracellular matrix from the germ tube (GT) of *Colletotrichum lindemuthianum*. Long fibers (fimbriae) projecting from the surface of the germ tube; (**B**) Ultrastructure of appressoria-like structures formed by *Pyricularia oryzae*. The melanin layer (arrows) is absent in the pore area (between arrowheads); (**C**) Close-up view of the rectangular section marked on Figure (**B**); (**D**) Fully developed appressorium of *Col. truncatum*. A: electron-dense extracellular matrix (arrowheads) coating on appressorium; DM: dialysis membrane; ECM: extracellular matrix; GT: germ tube; ML: melanin layer; N: nucleus; PC: parent conidium; PP: penetration peg; S: septum visible at the base of appressorium; V: vesicle. Scale bars: (**A**,**B**,**D**) = 1 μm. Re-drawn from (**A**): [67], (**B**): [23], (**C**,**D**): [69].

**Figure 4 pathogens-10-00746-f004:**
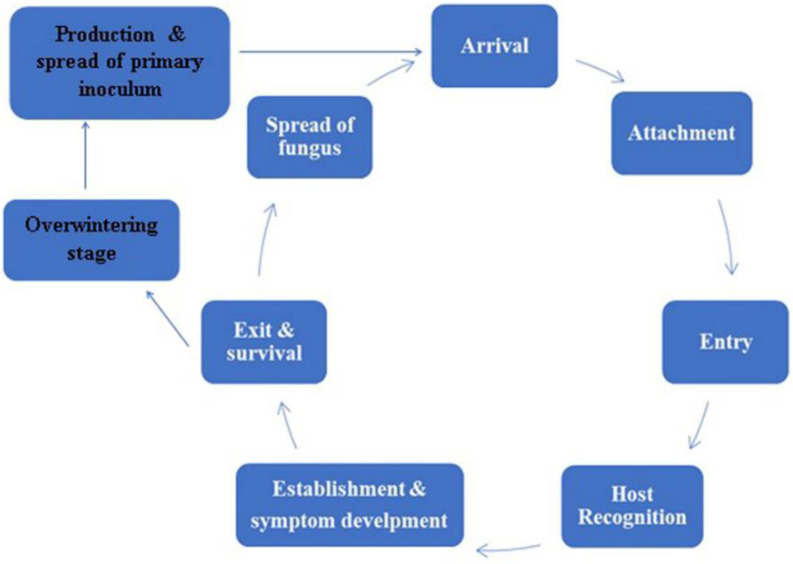
Events of a disease cycle.

**Figure 5 pathogens-10-00746-f005:**
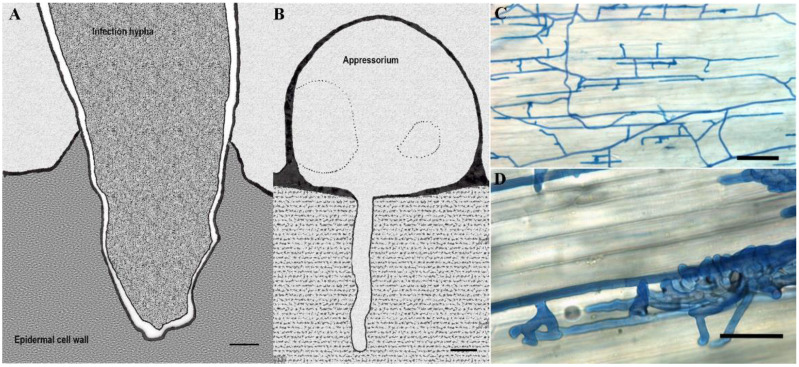
Different infection structures of phytopathogenic fungi. (**A**) An infection hypha of *Fusarium culmorum* invading the epidermal cell wall of the lemma at 36 h after inoculation (hai) observed transmission electron microscope; (**B**) Transmission electron micrograph of *Pyricularia grisea* appressorium formed on cellophane membrane with the penetration peg invading the substratum; (**C**,**D**) Light microscopy of compound appressoria development of *Rhizoctonia solani* on cauliflower hypocotyls after trypan blue staining of hyphae. Bulbous foot structures branch from runner hyphae on the plant (**C**). Lobate appressoria and infection cushions consist of agglomerated and highly ramified hyphae (**D**). Scale bars: (**A**,**B**) = 1 μm; (**C**,**D**) = 100 μm. Pictures re-illustrated from: (**A**): [83], (**B**): [3], (**C**,**D**): reprinted with permission from Pannecoucque, J.; Höfte, M. Interactions between cauliflower and *Rhizoctonia* anastomosis groups with different levels of pathogenicity. BMC Plant Biol. 2009, 9, 1–12 [84].

**Table 1 pathogens-10-00746-t001:** Different definitions for appressoria.

Definition
‘Peculiar spore-like organs produced by the germ tubes of spores of the bean anthracnose’ [43]
‘Localized swellings of the apices of germ tubes or older hyphae that develop in response to contact with the host’ [44]
‘The specialized cells, formed before the penetration of host tissue’ [45]
‘A swelling on a germ tube or hypha, especially for attachment in an early stage of infection’ [46]
‘Spore-like organs formed on germ tubes of *Colletotrichum lindemuthianum*, *Polystigma rubrum* and *Fusicladium tremulae*’ [9]
‘Hyphopodia are considered as attachment structures characteristic of a few families in three orders of fungi and appressoria as attachment organs characteristic of germ tubes or the early stages of infection’ [47] ‘Hyphal ramification within the host may be much more extensive from appressoria than from hyphopodia, since from the latter, a haustorium of only limited growth usually develops’ [47]
‘Appressoria can be defined as structures employed by fungal pathogens to press against and attach to the plant surface in preparation for infection’ [48]
‘Appressorium, a swelling on a germ-tube or hypha, especially for attachment in an early stage of infection, as in certain Pucciniales and in *Colletotrichum*; the expression of the genotype during the final phase of germination, whether or not morphologically differentiated from vegetative hyphae, as long as the structure adheres to and penetrates the host’ [49]
‘The organ of attachment of a germ tube or hypha of certain parasitic fungi in early stages of infection’ [50]
‘A flattened and thickened apex of a hyphal branch, formed by some parasitic fungi, that facilitates penetration of the host plant’ [51]

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
