# Peer review of "Diversity and Function of Appressoria"

_pathogens, 2021, doi:10.3390/pathogens10060746_

Round 1

Reviewer 1 Report

This review brings together older studies of appressorium morphology; reviews and data on ultrastructure and molecular genetics from the 1990s and early 2000s; and recent information on development and function of appressoria.

Infection structures in fungi are currently actively investigated, and it would be appropriate to re-check the primary literature and add information on recent studies of specific genes and their experimentally determined roles in appressorium function or on adhesion of the appressoria to the leaf surface.

  1. General definition of appressoria.
  • Line 92 Definition ambiguous, but seems to restrict appressoria to germ tubes.

  • Emmett and Parbery give a thoughtful definition.

"Alternatively, the scope of the concept could be expanded to incorporate all structures adhering to host surfaces to achieve penetration, regardless of mor­ phology. The only criteria delimiting an appressorium would then be its capacity to adhere to a host surface, and ability to germinate and penetrate the host. Thus the terms, appressorium and "infection structure," would become synonymous, and artificial restrictions imposed by morphology would be bypassed. It is in this latter sense that the term appressorium is used in this review."

  • Table 1. The following should be attributed to Frank not to Emmett and Parbery, who just quote it. It does not reflect the concept of Emmett and Parbery as the table suggests.

"Spore-like organs formed on germ tubes of Colletotrichum lindemuthianum, Polystigma rubrum and Fusicladium tremulae” [14]

  1. line 148. Does the cited ref [13] actually contain the description of B. cinera appressoria? Also, fix sp., cinerea.

  1. Section 7, Genomics, proteomics... provides broad generalizations and lists of citations without enough detail to allow readers to appreciate the interesting biology.

It would be more effective to incorporate relevant insights from genomics into sections on infection processes to keep the focus on biology rather than methods.

Whether reorganized or not, in revision, put more emphasis on examples illustrating biology, supported by just a couple relevant citations. Avoid long lists of species, genes, and references.

For example, in the following, it would be more interesting to have specific information about what the genes do:

  • line 485: "A plethora of infection-related gene families have been revealed through comparative genomic studies [40,111,112,118]...
  • line - 490 These expanded gene families principally help in the increased production of their encoding proteins and also provide the material for specific adaptations leading to evolution of new functional systems [31,41,118,120,121].

  • Line 515"Similar genomic and transcriptomic studies conducted on different plant and entomopathogenic fungi such as B. bassiana, B. graminis, Candida albicans, Cordyceps militaris, C. graminicola, C. higginsianum, Fusarium graminearum, Metarhizium anisopliae, M. acridum, P. grisea and Uromyces fabae functionally explored a repertoire of pathogenesis-related genes involved in surface attachment, appressorium formation and host penetration."

  • line 561 "revealed the functions of many genes involved in each step, from spore attachment to the release of effectors and other metabolites through penetration of hyphae into the 562 cell, leading to the establishment of many plant-fungal interactions 563 [5,17,24,28,30,33,34,36,37,39,42,82,90,92,98,103,110]."
  •  
  1. Line 323, Text starting here. Check sentence structure through next paragraphs. Purpose of paragraphs is unclear. Make sure that each paragraph makes a coherent argument beginning with a topic sentence. Meaning of sentences is too often unclear.

  • Line 327. Is 'chitin binding secreted by fungus' correct or should it be by the plant? "Chitin binding proteins that are secreted by the fungus also facilitate both germ tube de- 327 velopment and the initiation of contact on the hydrophobic surface in P. oryzae"

  • Line 343. What are 'cutinase reactions'? Also, clarify, fatty acids aren't formed as a result of phenols, volatiles are they? Clarify meaning in 'These inducers include fatty acids that are formed as a result of cutinase reactions, phenols and volatiles'
  •  
  • Line 352. Missing words? Edit for clarity: "Furthermore, gene characterization research provides ample evidence that transcriptional factors that are crucial for activating genes in response to plant defense stimuli [28,36,39]."

  • Line - 351 Clarify, why is MAPK an exception? 'With the exception of MAPK, both cAMP and calcium signalling cascades, regulate appressoria development and pathogenicity in fungi such as P. oryzae [33] and Colletotrichum species [90]'

  1. Line 392. Incorrect, remove: "Appressorial pore is the point of infection, which lacks the cell wall, facilitating the contact of the fungal plasma membrane with the plant surface."

  • Penetration pegs do have cell walls. The walls are not melanized and are not stained in freeze substituted material as in the work by Dyke and Mims (who were careful to say 'wall appears to be absent'), but other studies using markers show them to be present, eg Bourett & Howard 1990 used TEM, WGA. https://cdnsciencepub.com/doi/abs/10.1139/b90-044. Walls may be modified so chitin can't be detected in them. But walls are present.

Figures

  1. Figure one and Line 146. "we grouped into compound appressoria category as in Emmett and Parbery [14]". Is this true? Emmett and Parbery included a broader range of types of appressoria in the compound group where this paper does not. E & P's classification makes sense given the diversity of forms on and in plant tissue.

  1. 2. How is each type of appressorium to be classified? Specify in figure legend.

  1. 3 Permissions for 3A? The wall in the freeze sub cell's penetration peg is not stained, but that does not mean that it is not present.

  1. Figure 6. Integrate the figure more fully with text or delete it. The current version is confusing. It is cited only in general support of pathogen processes, not used in a meaningful way to support understanding of text. Processes within cells are confused with extra-cellular processes. Hard to read labels because text is small and fuzzy. Legend gives insufficient explanation.

If Fig. 6 is to be retained, individual cartoons should be labelled 'a, b, c' and each part should be cited in the section of text where the cartoon would be helpful. The legend should quote the text in describing each cartoon.

Typos and minor corrections

  1. Unless it conflicts with journal policy, italicize names of genes like Pmk1, 347 MEKK-Mst11, Osm1. This helps distinguish genes from proteins.

  1. If the name of the gene contains the initials of the species name, specify the species.

  1. Genus and species names, italicize throughout.

  1. Correct; currently links to erratum, not manuscript: 23. Inoue, K.; Suzuki, T.; Ikeda, K.; Jiang, S.; Hosogi, N.; Hyon, G.S.; Hida, S.; Yamada, T.; Park, P. Extracellular matrix of Mag- 672 naporthe oryzae may have a role in host adhesion during fungal penetration and is digested by matrix metalloproteinases. J. Gen. 673 Plant Pathol. 2008, 74, 96–96. https://doi.org/10.1007/s10327-007-0071-3

  1. Line 166 Subject is 'ultrastructural studies' so this line needs editing to make sense:'Therefore, they provide notable targets for developing control measures for plant diseases. '

  1. Line 162. Abbreviation of genera is inconsistent. Consider avoiding abbreviating genus where many genera have recently been mentioned that start with the same letter. Here, for example, which 'P' is meant? C. lindemuthianum and P. oryzae on sterilized dialysis membranes.

Spell check. For example

  1. cellulaases [104] 426

  1. Pyricularia grisea Cutinase2 genes, lower case for 'C'

  1. Line 505. Assuming the gene codes for mRNA, it is involved in protein synthesis, so clarify which proteins are meant: 'In contrast, the genes with significantly decreased expression were those that were involved in protein synthesis.'
  2.  
  3. Text can be more concise in several places, for example

line 508. From: According to the transcriptome analysis, it was found that number of genes related to protein metabolism were differentially expressed during appressorium formation, such as SPM1 invloved with penetration, and Mgd1, GAS1, GAS2 related to appressorial formation....

To:

Transcriptome analysis showed differential expression of Mgd1, GAS1, and GAS2, genes that are related to appressorium formation, and of SPM1, involved in host penetration

Author Response

Dear Reviewer,

Thank you

Reviewer 2 Report

The paper contain a very good review about apppressoria

Here are some observations and recommendations:

  • Row 190-Fig.3-D- ECM and Ml are not found
  • Rows 198 and 199 – which is authors’ oppinion about the mechanisms of melanin layer in adhesion?
  • Row 201- Golgi bodies and Woronin bodies
  • Rows 249 and 250 contain the information and reference similar to rows 252 and 253
  • Row 329 germtube, germ tube acts

Author Response

Dear Reviewer,

Thank you.
